# Mice and Habitat Complexity Attract Carnivorans to Recently Burnt Forests

**Roger Puig-Gironès *** and **Pere Pons**

Departament de Ciències Ambientals, University of Girona, 17003 Girona, Catalonia, Spain; pere.pons@udg.edu
*   Correspondence: roger.puig.girones@gmail.com

**Abstract:** Faunal responses to wildfire depend on the fire effects on direct mortality, habitat structure, and resource availability for animals. Despite the importance of large predators in terrestrial trophic webs, little is still known about how fire affects carnivorans (the mammalian order Carnivora). To evaluate the responses of the carnivoran community to fire, we studied three recently burnt forest areas in the western Mediterranean basin. Line transects were used to quantify evidence of carnivorans (mainly feces) and to measure environmental variables and resources (small mammal abundance, fleshy fruit availability, and plant cover). Throughout the study, we found 212 carnivoran field signs, 93% of them produced by red fox and stone marten. Immediately after fire, carnivoran occurrence was more frequent close to the perimeter of the burnt area, where fire severity was low, and in places with greater small mammal abundance. Small mammal abundance and plant cover had the greatest effect on the frequency of occurrence of red fox in the burnt area surroundings, and this increased with time-since-fire in the burnt area. Furthermore, the presence of red fox did not affect stone marten occurrence. Stone martens were found around the burnt area perimeter, probably because of their preference for high plant cover, and they were not significantly affected by small mammal abundance. The scat frequency of occurrence of both species was not significantly related to fleshy fruit availability. Accordingly, rodents and carnivorans were more abundant where the habitat was more complex. Our results show that the responses of some carnivorans to fire are influenced, directly and indirectly, by habitat structure and resource availability.

**Keywords:** food availability; Mediterranean ecosystems; predator–prey interaction; red fox; small-mammals; stone marten; wildfire

## 1. Introduction

Wildfires have broad effects on ecosystems, and are important drivers of biome distribution and the composition of biotic communities [1–3]. Faunal responses to wildfires are influenced by fire effects on habitat structure and resources, direct mortality, and an increase in predatory pressure [4,5]. Furthermore, wildfires can represent an opportunity for certain opportunistic, generalist, or open-habitat animal species [6–8].

The capacity to predict how fauna responds to wildfires is fundamental in promoting its conservation [9]. Nevertheless, this capacity needs to be enhanced, mainly because of a lack of knowledge about the interactions between fire and other drivers of species distributions [10,11]. Predator–prey relationships are a potential driver because predators influence the structure and function of prey communities and vice versa [12]. Mammalian carnivores are mesopredators and top predators in terrestrial ecosystems around the world. Nevertheless, how fire affects carnivorans (members of the mammalian order *Carnivora*) and their biotic and abiotic interactions is still largely unknown, especially in fire-prone Mediterranean-basin ecosystems [9].

Carnivorans have large home ranges, which can include a wide variety of habitats, and occur at low densities, making it difficult to detect their responses to fire. In a recent review, Geary et al. [9] highlighted a lack of a clear response by carnivorans to fire, varying according to locations, habitat types, or species assemblages. In the Mediterranean basin, under intensive fire regimes, wolves seem resilient to fire [13]. On the other hand, the red fox (*Vulpes vulpes* Linnaeus, 1758) was described as a regular visitor in a burnt area in Macedonia, while the occurrence of stone martens (*Martes foina* Erxleben, 1777) was recorded in the surroundings of the burnt area perimeter [14]. Both species may also present interspecific competition and trophic interactions [15]. The responses of red fox and stone marten to fire may be related to their habitat structure preferences, intra-guild interactions [9], plasticity to take advantage of a changing environment, and generalist diets [8]. Red fox scats found near to the burnt area perimeter in another Mediterranean burnt area contained seeds, but scats within the burnt area did not, since there were not enough fleshy fruits and the red fox may have found other resources [16]. Hence, carnivoran presence and abundance may be affected by food resource availability in burnt areas [17,18].

After fire, the density of small mammals can be higher in burnt than in unburnt areas [19,20] and, especially, in the surroundings of the burnt area perimeter, where cover is greater and habitat structure more complex compared to the inner burnt area [21]. Burnt areas can also support increasing abundance of small mammals through time along with plant cover recovery. However, plant recovery varies due to differences in soil, topography, rainfall, fire severity, and plant community [22,23]. Consequently, predators may be attracted to recently burnt areas if prey is abundant or more easily captured [24]. However, if prey abundance is low or if predators are unable to hunt effectively with low plant cover, predators may avoid recently burnt areas [25,26]. Therefore, post-fire habitat structure may affect predator–prey interactions by modifying encounter rates, available refuges [27,28] as well as affecting prey behavior, abundance, or distribution [29,30].

In this study, we aimed to evaluate the carnivoran response to fire in relation to habitat structure, fire severity, resource availability (small mammals and fleshy fruits), and carnivoran species segregation in three recently burnt forests in Catalonia (western Mediterranean basin). We predicted that (1) small mammal abundance and fleshy fruit availability affect carnivoran abundance in recently burnt areas; (2) immediately after fire, carnivoran abundance will be highest in the surroundings of the burnt area perimeter, and (3) spatial segregation between the stone marten and red fox affects their distribution in the burnt area.

## 2. Material and Methods

### 2.1. Study Area

Three recently (less than six months) and large (over 200 ha) severely burnt areas (i.e., >80% of immediate tree mortality within the fire perimeter) distributed throughout Catalonia (NE Iberian Peninsula) were studied. Two of our study areas were located in lowlands (90 to 574 m) under a Mediterranean climate (La Jonquera and Rasquera), whereas the other was in the Pyrenees (1461 to 1890 m) with a humid mountain climate (Ger). The La Jonquera study area was burnt in July 2012 by a wildfire affecting 13,088 Ha of oak, pine, and shrubland habitats. Nine transects were sampled on 15 occasions, starting two weeks after fire. The Rasquera study area was burnt in May 2012 by a wildfire affecting 3082 Ha of pine and shrubland habitats. In this case, five transects were sampled seven times, starting 18 weeks after fire, because it occurred 12 and 16 weeks before the La Jonquera and Ger fires, respectively. The Ger study area was burnt in August 2012 by a wildfire affecting 250 Ha of pine and shrubland habitats. Here, eleven transects were sampled on 16 occasions, starting one week after fire. All sites were studied for 30 months (for more information about study areas, see [21,31]).

## 2.2. Focal Species

We focused on the carnivoran community, composed of red fox, stone marten, genet (*Genetta genetta* Linnaeus, 1758), European badger (*Meles meles* Linnaeus, 1758) and least weasel (*Mustela nivalis* Linnaeus, 1766); red fox and stone marten being the most common species in the three areas studied. The red fox is a territorial species with great ecological plasticity. Small mammals, lagomorphs, and fleshy fruits represent a large part of the biomass consumed by red fox in the Iberian Peninsula [32]. The stone marten is also a territorial species that prefers areas with developed tree cover [33]. Its abundance may be related to fleshy fruit plants such as strawberry trees (*Arbutus unedo* L.) [34]. Its diet comprises a broad spectrum of items and it presents adaptation to local prey availability, the main foods ingested being small mammals and fleshy fruits [35,36]. The competition and trophic interaction with the red fox may lead the stone marten to exploit trophic resources that the red fox cannot exploit such as shrub fruits [15,37]. Both species may also act as seed dispersers; however, stone martens disperse seeds more effectively in sites with abundant and continuous forest cover [38].

## 2.3. Sampling Design and Field Methods

We established line transects (approximately 900 m in length and 6 m wide) perpendicular to the burnt area perimeter within each study area, in which environmental and resource (food and refuge) variables were measured and evidence of carnivorans (hereafter field signs) was recorded. Transects had approximately 100 m in the unburnt area. Adjacent transects were separated by at least 140 m in its closest point, although 61% of adjacent transects were separated by at least 500 m. The average distance between adjacent transects of 2048 m ensured a reasonable spatial independence of transects for carnivorans and to avoid double counts. Moreover, transects were located across straight borders of the burnt area to avoid the influence of border geometry on edge effects [39]. La Jonquera concentrated 22 of the 33 transects analyzed, while Ger had six and Rasquera five. An observer (the same for all transects and study areas) walked each transect at low speed, looking for evidence of carnivorans (Figure 1) within a 3 m width at each side of the transect line. For each field sign, the observer noted the type (feces, direct observation, latrines, footprint), determined the species, the UTM coordinates, and removed the feces from the transect and/or erased the footprints to avoid counting it on subsequent occasions. Evidence of five species was found (Table 1). However, field signs did not allow carnivoran abundance to be estimated due to the heterogeneity in species detectability, derived from, among others, particular habits of marking the territory. Species like the red fox and the stone marten deposit their feces in clearly visible places. Others, like the common genet, often use latrines to defecate, so feces presents an aggregate distribution, which is more difficult to detect. Both animal weight and substrate type determine species detection through footprints and, in general, it is easier to detect heavier animals such as the red fox or European badger than smaller, lighter animals such as the least weasel. The least weasel is the only species in the area that can be captured by Sherman traps. Therefore, field signs were used as a surrogate of carnivoran frequency of occurrence at the species level and for the whole community.

Given that carnivorans have great territorial mobility and low density [9], each transect was divided into categories of *distance from the burnt area perimeter*. Negative values were assigned to stations in the unburnt area (−1 = 1 to 50 m; −2 = 51 to 100 m; and −3 = more than −101 m from the burnt area perimeter), while positive values indicated distances within the burnt area (0 = 0 m; 1 = 1 to 50 m; 2 = 51 to 100 m; 3 = 101 to 200 m; 4 = 201 to 400 m; 5 = 401 to 800 m; and 6 = more than 801 m from the perimeter). *Time-since-fire* was also grouped in monthly categories (1 = 1 to 2 months after fire; 2 = 3 to 4; 3 = 5 to 6; 4 = 7 to 10; 5 = 11 to 15; 6 = 16 to 20; 7 = 21 to 25; and 8 = 26 months to end of sampling). In La Jonquera and Ger, sampling frequency was monthly in the first six months, bimonthly from seven to 24 months, and every three months thereafter until 30 months had elapsed since the fire. The Rasquera site was sampled with a three-month frequency from the beginning of the sampling because sampling began 18 weeks after the fire.

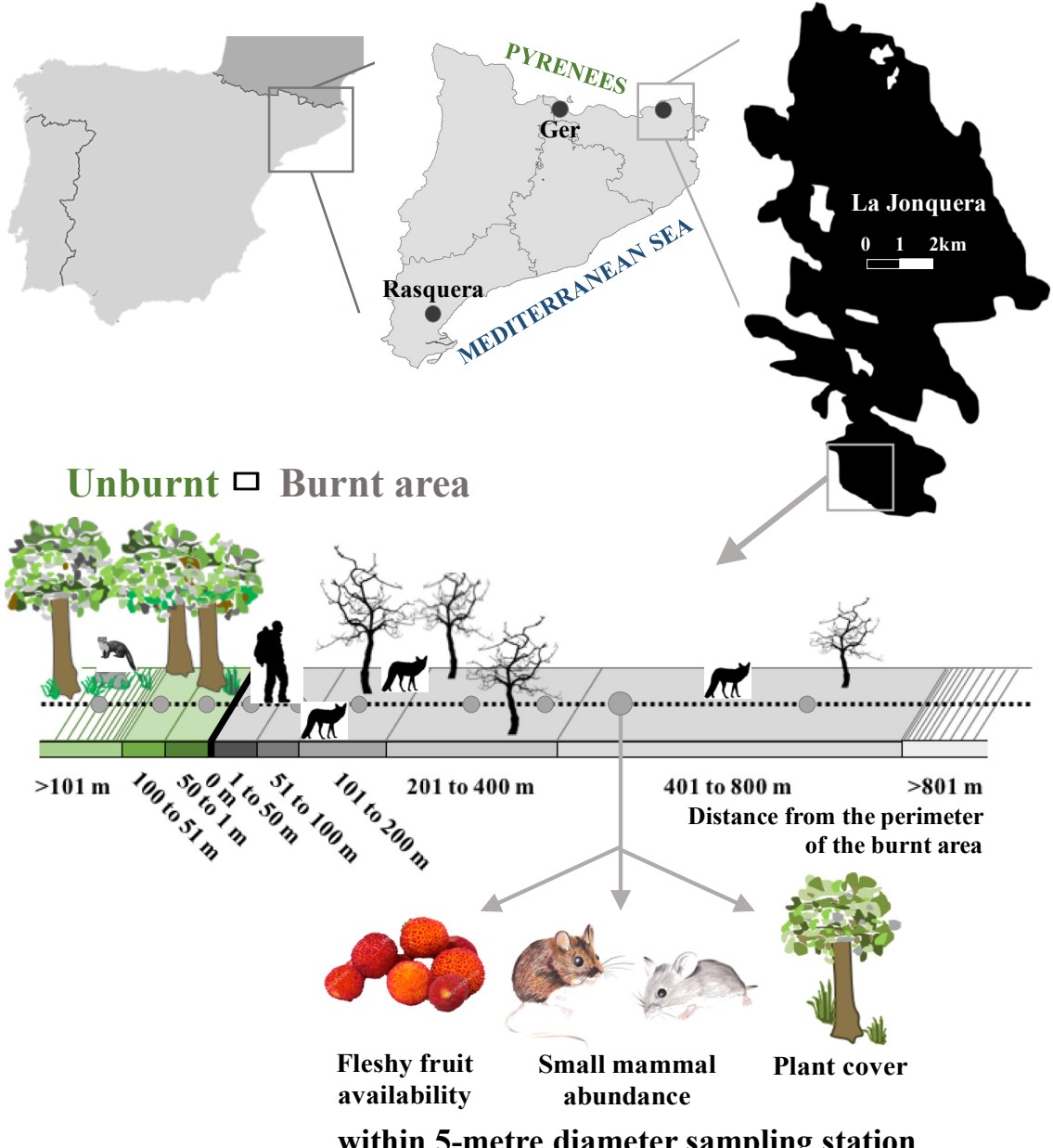

**Figure 1.** Location and schematic representation of the sampling methods. Location of the three study areas in Catalonia (NE Iberian Peninsula), using the La Jonquera burnt area as an example in black [40], and schematic representation of linear transects, perpendicular to the burnt area perimeter with sampling stations (grey dots) at increasing distances from the burnt area perimeter. In each sampling station of 5 m diameter, we measured vegetation structure, production of trophic resources, and relative abundance of small-mammals (with a Sherman live trap).

In each transect, sampling stations of 5 m in diameter were established, in which we measured relative small mammal abundance, fleshy fruit availability, and other environmental variables (Figure 1). The exact distance from the nearest point of the burnt area perimeter to each sampling station was calculated. Sherman live traps for small mammals (5.1 × 6.4 × 22.3 cm, Sherman, Tallahassee, FL, USA) were set in the center of each sampling station and were baited with a mixture of tuna, flour, oil, and a piece of apple [19] and provided with thermal protection. The small mammals caught were identified to species, sexed, and marked with ear tags (National Band Co., NewPort, KY, USA). All sampling procedures met the American Society of Mammalogists (ASM) Care and Use guidelines [41]. A sampling occasion was defined as the three consecutive days of capture per station.

Sherman traps were checked early in the morning daily for the three days following their installation. We live-trapped four species of small mammals: 66.8% of which were wood mice (*Apodemus sylvaticus* Linnaeus, 1758); 18.1% Algerian mice (*Mus spretus* Lataste, 1883); 10.6% greater white-toothed shrew (*Crocidura russula* Hermann, 1780); and 4.5% common vole (*Microtus arvalis* Pallas1778). We assumed that the number of individuals captured would increase with overall small mammal abundance around sampling stations [42,43]. This sampling assumption is adequate with low small mammal abundance, as in our case, where only 12 out of 1400 sampling occasions among all stations (i.e., 0.9%) corresponded to the maximum of three captures. On the other hand, 481 and 98 stations (i.e., 34.4 and 7%) produced one and two captures per occasion, respectively.

**Table 1.** Number of carnivoran evidences per type.

| | Species | Carnivoran Community | Red Fox (*Vulpes Vulpes*) | Stone Marten (*Martes Foina*) | Common Genet (*Genetta Genetta*) | Least Weasel (*Mustela Nivalis*) | European Badger (*Males Meles*) |
|---|---|---|---|---|---|---|---|
| | Feces | 189 | 103 | 85 | | | 1 |
| Number per Type | Active Latrines | 10 | | | 10 | | |
| | Footprints | 4 | 3 | | | | 1 |
| | Captures | 4 | | | | 4 | |
| | Observations | 5 | 5 | | | | |
| **Number of Total Evidences** | | 212 | 111 | 85 | 10 | 4 | 2 |

Number of total and single carnivoran species found in the three burnt areas studied segregated into types of evidence for the 2.5 years sampled.

*Fleshy fruit availability* at sampling stations was estimated from the total volume of each shrub inside the 5 m diameter station and the number of counted fruits at three random branches of known volume. Volumes were calculated using the formula of the cone volume ($V = \frac{1}{3}\pi r^2 \cdot h$), where $r$ is the radius and $h$ is the cone height (for more information, see [21]). Fleshy fruit abundance was estimated mainly from *Amelanchier ovalis*, *Arbutus unedo*, *Arctostaphylos uva-ursi*, *Chamaerops humilis*, *Coriaria myrtifolia*, *Crataegus monogyna*, *Juniperus communis*, *J. oxycedrus*, *J. phoenicea*, *Myrtus communis*, *Pistacia lentiscus*, *Prunus spinosa*, *Rhamnus alaternus*, *Ribes rubrum*, *Rosa* sp., *Rubus* sp., *Sambucus nigra*, *Smilax aspera*, *Solanum nigrum*, *Sorbus aria*, *S. aucuparia*, and *Vaccinium myrtillus*.

Habitat structure was characterized by live foliage cover (in %) and estimated by comparison with a reference chart [44] for six virtual vegetation height layers: 0–0.25 m (C0), 0.25–0.5 m (C25), 0.50–1 m (C50), 1–2 m (C100), 2–4 m (C200), and more than 4 m (C400). A principal component analysis (PCA) was then used to summarize the information obtained from the six vegetation layers, after arcsine-transforming cover values. The first component (PC1, *plant cover*, with 42.6% of explained variance) corresponded to the magnitude of plant cover, mainly understory, and the second (PC2, *height of vegetation,* with a 27.9% of explained variance) ordered stations in terms of the maximal height of vegetation (Figure S1). We further characterized the possible role of potential fire refuges for fauna, measuring their distance to sampling stations in the burnt area (*distance from closest refuge*). We considered refuges as unburnt vegetation patches of at least 20 m², rock outcrops or heaps, water points (ponds, creek, etc.), rural constructions (dry-stone walls, farmhouses, barns, etc.), and field crops (both active and abandoned). This metric distance was transformed using the square root. We also estimated *fire severity* using a categorical scale following the methodology described by Keeley [45]: (1) unburned; (2) unburned but plants exhibit leaf loss from radiated heat; (3) canopy trees with green needles and herbs charred; (4) moderate or severe surface burn; and (5) deep burning or crown fire. Fire severity is defined by the impact that a fire has on the ecosystem and assesses the changes induced by the fire in vegetation and soil properties [45].

*2.4. Statistical Analyses*

Generalized linear mixed models (GLMM) with a Poisson error structure and logit link function were used to analyze the effects of food availability (small mammals and fleshy fruits), environmental, and temporal variables on the carnivoran community. *Transect* nested within *site* (i.e., the three study areas) and *climatic region* (i.e., Mediterranean or Pyrenees) was included as a random factor in order to control possible site-based differences and temporal pseudoreplication. We assessed the importance of variables derived from resource availability (*small mammal abundance* and *fleshy fruit availability*), variables related to fire (*time-since-fire, distance from the burnt area perimeter, time-distance interaction* and *fire severity*), and variables describing habitat structure (*distance from closest refuge, plant cover, height of vegetation*) in the variability of carnivoran frequency of occurrence. To explore the effect of food availability, environmental, and temporal variables on the red fox and stone marten occurrence, we used the same analysis structure described for the carnivoran community. For the stone marten analysis, we also used red fox occurrence to evaluate possible exclusive competition.

An information-theoretic framework (IT) was used to obtain the final model for each dependent variable (carnivorans, red fox and stone marten occurrence). Model selection was based on the Akaike Information Criteria corrected for small samples (AICc) [46] and followed a series of hierarchical steps. (1) Multicollinearity diagnostics [47] were performed by quantifying variance-inflation factors (VIF), and generalized variance-inflation factors (GVIF [1/(2d.f.)] with several categorical predictor variables with multiple degrees of freedom) were calculated for each fixed factor [48], where large VIF or GVIF values (arbitrary threshold of ≤2.5 suggesting collinearity) were sequentially dropped from further analysis [49]. (2) From the set of variables selected, all the possible combinations of the predictor variables (described in previous paragraphs), generating different biologically meaningful models, were explored. (3) Normality and homoscedasticity of residuals were checked by visually inspecting the plots of residuals against fitted values. (4) For each model, the AIC weight (AICω) was calculated (total AICω adds 1) [50]. Furthermore, if there was no clear most parsimonious model (one or more models showed a difference in AIC of less than two from the best model), we proceeded to estimate the average final model from all of those models [46]. (5) To interpret the magnitude of each variable on the average final model, the AIC weight (ω+) was calculated (Table S1). Furthermore, if standard errors (SE) were large (1.96 * SE > parameter appreciation, for the 95% confidence intervals) the estimate of the parameter was considered imprecise. To perform these analyses, we used R software [51] with the lme4 [52], MuMIn [53] package. After performing multicollinearity diagnostics and since distance from the burnt area perimeter and fire severity were correlated, we eliminated fire severity from all the analyses, assuming that distance acts as a substitute for fire severity.

Although the GLMMs above tested our hypothesized effects, it was also possible that carnivoran frequency of occurrence responded to another variable or had been affected by complex indirect effects. Therefore, to control these possible confounding variables, we used structural equation modeling (SEM) performed using the R package Lavaan [54]. Path analysis was used to analyze the direct and indirect effects of food availability, environmental, and temporal variables on the carnivoran community. Time-since-fire, distance from the burnt area perimeter, pre-fire habitat, and vegetation structure may have both direct and indirect effects on the carnivorans. To test this complex system, we constructed an initial (maximal) model containing all possible biologically meaningful pathways able to affect, directly or indirectly, carnivoran occurrence. We then simplified the model down to a final model (lowest AIC score), and the initial and final simplified models are presented in Figure 2 (see Text S1 and Table S2 for detailed information).

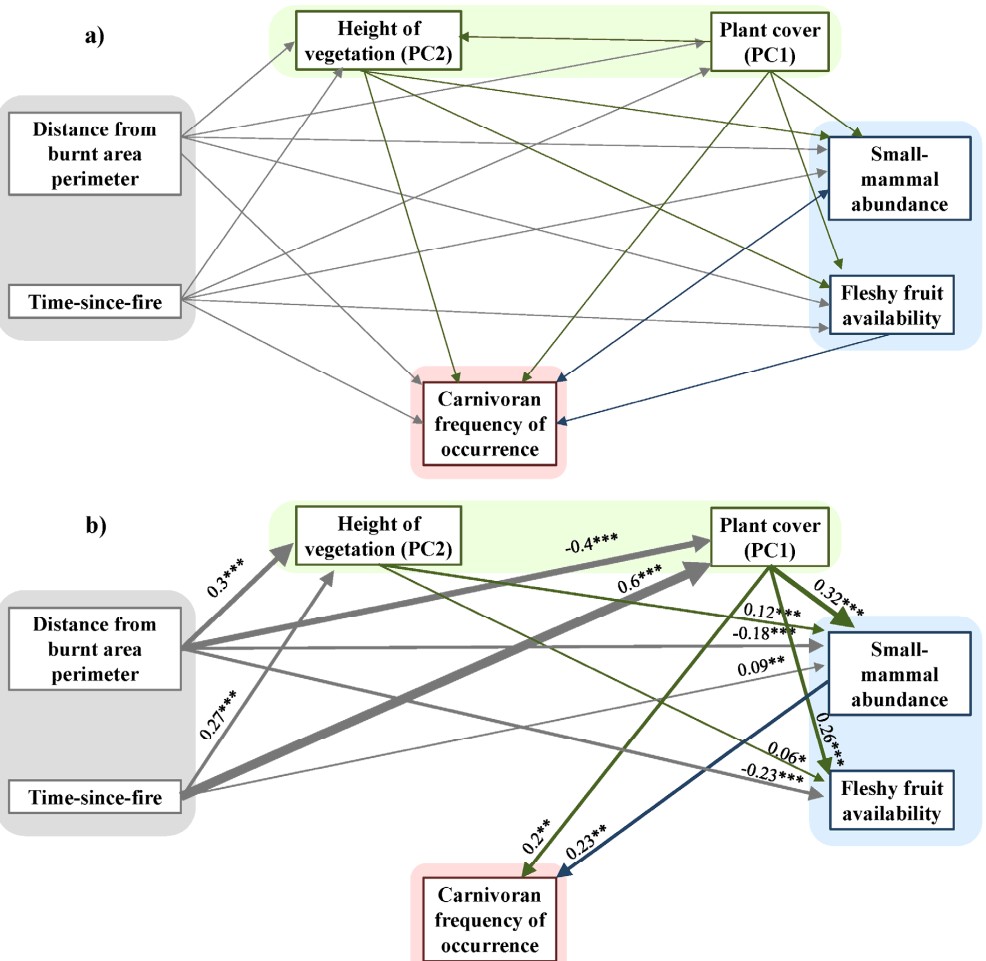

**Figure 2.** Most parsimonious structural equation models (SEM). Functions shown are carnivoran frequency of occurrence. All paths in the initial models (graph (**a**)) were treated as optional, and were thus able to be removed during model simplification (from 27 to 22 parameters) using maximum likelihood estimation. As a final model (graph (**b**)), we selected the model with the lowest AIC score. Values adjacent to paths indicate standardized direct coefficients, with significance indicated by * $p < 0.05$; ** $p < 0.01$; *** $p < 0.001$. The model was constructed using four hierarchical packages: (1) time and distance in relation to fire, (2) habitat structure (first two PCA axis), (3) food availability, and (4) carnivoran occurrence. One-way arrows represent paths between variables.

## 3. Results

Throughout the study, we found 212 field signs of carnivorans including 52.4% belonging to red fox, 40.1% to stone marten, 4.7% to common genet, 1.9% to least weasel, and 0.9% to European badger. In total, 198 field signs were feces (89.2%), 10 were active latrines (4.7%), five direct observations (2.4%), four were captures (1.9%), and four were footprints (1.9%). All transects had field signs (range 2–21 for the duration of the study), although four transects concentrated 41% of the field signs found.

Generalized linear mixed models (GLMM) showed that the time—distance interaction affected the presence of carnivorans, with greater frequency of occurrence found close to the perimeter immediately after fire, while occurrence increased within the burnt area with time-since-fire (Figure 3a). Both variables, *time-since-fire* and *distance from the burnt area perimeter*, were related to habitat structure complexity, because fire severity was lower near the burnt area perimeter, and the recovery of small mammals in burnt areas mainly started near its perimeter. With respect to food availability variables, carnivoran occurrence showed a positive relationship with small mammal abundance. Although, fleshy fruit availability seemed to be an influencing variable, it was not significant (Table 2).

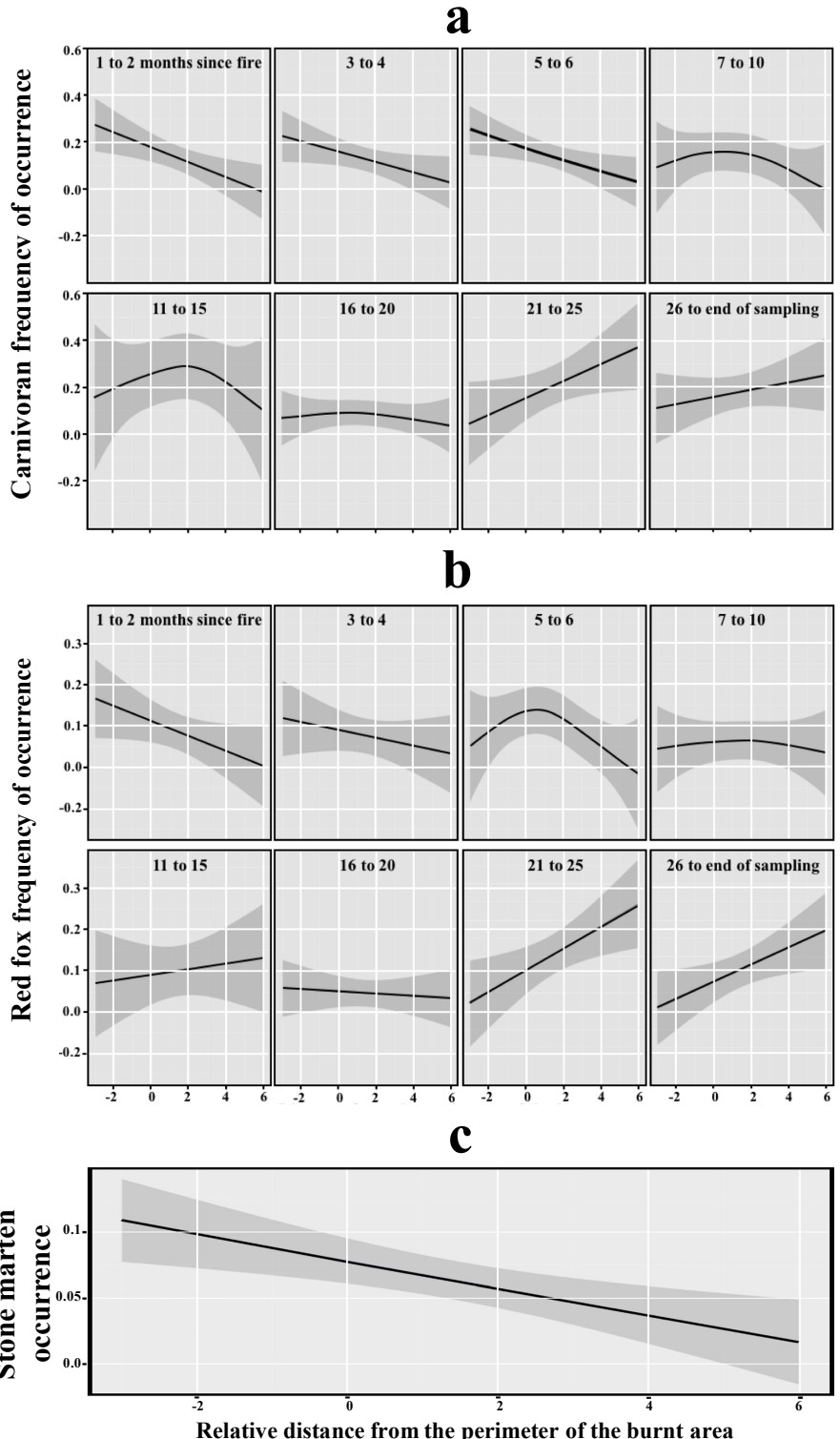

**Figure 3.** Model predictions of carnivoran frequency of occurrence over time and distance. Marginal effects measure the instantaneous rate of change of the model predictors on (**a**) carnivorans, (**b**) red fox, and (**c**) stone marten frequency of occurrence according to time-since-fire categories and the relative distance from the perimeter of the burnt area. For the distance variable, negative values were assigned to stations in the unburnt area (−1 = 1 to 50 m; −2 = 51 to 100 m; and −3 = more than −101 m from the burnt area perimeter), whereas positive values indicate distances within the burnt area (0 = 0 m; 1 = 1 to 50 m; 2 = 51 to 100 m; 3 = 101 to 200 m; 4 = 201 to 400 m; 5 = 401 to 800 m; and 6 = more than 801 m from the burnt area perimeter). Trend line and standard error shown were obtained from GLMM model estimates.

**Table 2.** Effect of environmental variables on carnivoran frequency of occurrence.

| Explanatory Variables | Carnivoran Community | | | Red Fox (*Vulpes vulpes*) | | | Stone Marten (*Martes foina*) | | |
|---|---|---|---|---|---|---|---|---|---|
| | Coefficient ± SE | $p$ | ω+ | Coefficient ± SE | $p$ | ω+ | Coefficient ± SE | $p$ | ω+ |
| *Intercept* | −1.96 ± 0.21 | *** | | −2.25 ± 0.24 | *** | | −2.7 ± 0.29 | *** | |
| Time-since-fire | −0.08 ± 0.04 | * | 1.0 | −0.25 ± 0.06 | *** | 1.0 | | | |
| Distance from the burnt area perimeter | −0.29 ± 0.07 | *** | 1.0 | −0.26 ± 0.09 | ** | 1.0 | −0.16 ± 0.06 | ** | 1.0 |
| Time-distance interaction | 0.06 ± 0.01 | *** | 1.0 | 0.08 ± 0.02 | *** | 1.0 | | | |
| Distance from closest refuge | | | | | | | −0.06 ± 0.07 | ns | 0.17 |
| Small-mammal abundance | 1.46 ± 0.35 | *** | 1.0 | 2.26 ± 0.47 | *** | 1.0 | 0.82 ± 0.58 | ns | 0.49 |
| Fleshy fruit availability | 0.02 ± 0.01 | ns | 0.53 | 0.008 ± 0.02 | ns | 0.21 | | | |
| Plant cover (PC1) | | | | | | | 0.28 ± 0.08 | ** | 1.0 |
| Height of vegetation (PC2) | | | | | | | 0.14 ± 0.11 | ns | 0.37 |
| Red fox frequency of occurrence | | | | | | | 0.25 ± 0.25 | ns | 0.30 |

Summary of the selected models, derived from generalized linear mixed models (GLMM) analyses, of total carnivoran species, red fox, and stone marten occurrence. The table shows the model parameter coefficient and its standard error (±SE), the associated $p$-values ($p$: ns = non-significant; * $p < 0.05$; ** $p < 0.01$; *** $p < 0.001$), and the Akaike Information Criteria weight (AICω+). *Intercept* is the value of carnivoran occurrence when all the covariates are = 0, while $p$-value indicates whether it is significantly different from 0.

Small mammal abundance had the largest standardized total effect on carnivoran frequency of occurrence, with plant cover having the second largest effect in the structural equation model (SEM) (Figure 2). The effects of distance from the burnt area perimeter and time-since-fire on carnivorans (Figure 3a) were indirect, mediated by plant cover and small mammals (Table 3). Fleshy fruit availability had no significant influence on carnivoran occurrence. On the other hand, distance from the burnt area perimeter, time-since-fire, plant cover, and height of vegetation significantly affected small mammal abundance. Distance from the burnt area perimeter and time-since-fire also indirectly affected small mammals via plant cover. While distance from the burnt area perimeter, plant cover and height of vegetation directly influenced fleshy fruit availability, we also found indirect effects via plant cover of both the distance from the burnt area perimeter and time-since-fire. Both components of habitat structure (plant cover and height of vegetation) were directly affected by distance from the burnt area perimeter and time-since-fire.

**Table 3.** Structural equation models (SEM) indirect standardized effects.

| Response Variable | Explanatory Variables | Standardized Effects | $p$ |
|---|---|---|---|
| **Flesh fruits availability** | Distance from the burnt area perimeter → Plant cover | −0.10 | *** |
| | Time-Since-Fire → Plant cover | 0.16 | *** |
| **Small-mammal abundance** | Distance from the burnt area perimeter → Plant cover | −0.12 | *** |
| | Time-Since-Fire → Plant cover | 0.19 | *** |
| | Distance from the burnt area perimeter → Plant cover | −0.08 | ** |
| **Carnivoran occurrence** | Distance from the burnt area perimeter → Small-mammals | −0.03 | * |
| | Time-Since-Fire → Plant cover | 0.12 | ** |

Structural equation modelling (SEM) indirect standardized effects and its associated *p*-values (*p*: ns = non-significant; * *p* < 0.05; ** *p* < 0.01; *** *p* < 0.001) for all effects tested on final model of the total carnivoran species.

The red fox followed similar trends to those described for the whole carnivoran community and showed greater frequency of occurrence near the burnt area perimeter immediately after fire. However, this species expanded to the inner burnt area and reached greatest occurrence there 21 months after the fire (Figure 3b). Small mammal abundance influenced red fox occurrence as well as fleshy fruit availability and pre-fire shrubland habitat (Table 2). Stone marten frequency of occurrence was not influenced by time–distance interaction or time-since-fire. However, the distance from the burnt area perimeter negatively affected stone marten occurrence (Figure 3c). Furthermore, both a pre-fire forest habitat and greater plant cover positively affected its occurrence. Abundance of small mammals, height of vegetation, and red fox occurrence also had a positive influence on stone marten occurrence (although not significant, they were present in the selected model), while distance from closest refuge showed negative effects. Fleshy fruit availability did not seem to affect this species (Table 2).

## 4. Discussion

Our results showed that small mammal abundance affected carnivoran frequency of occurrence in recently burnt areas. However, fleshy fruit availability did not seem to be a determining variable for their presence. Similarly, red fox increased its presence in sites with greater small mammal abundance and was not significantly affected by fleshy fruit availability, in accordance with its usual relevance (approximately 15% of the frequency of occurrence on average in Spain) in the diet of the species [32,55]. Then, the results partially support the hypothesis that small mammal abundance and fleshy fruit availability affect carnivoran abundance in recently burnt areas. In burnt areas distributed throughout Catalonia, we previously reported that rodent foraging activity increased considerably within the burnt area 15 months after fire, regardless of the fleshy fruit and seed availability [21,31]. This pattern may be behind the increase in red fox occurrence within the burnt area 21 months after fire [16]. In contrast, stone marten occurrence was not significantly affected by small mammals, probably due to its rich diet in fruits and insects [34,56,57]. Therefore, higher carnivoran occurrence in recently burnt areas is related to greater prey availability [21] and the loss of plant cover used as shelter [58].

Post-fire vegetation structure and its succession may create variable predation environments in space and time. Likewise, this relationship may influence prey abundance and its effects on ecosystem components and functions [19]. Accordingly, post-fire habitat structure and, likely, the perception of predator risk affects prey behavior [29,30]. Moreover, habitat structure draw predators into burnt areas if prey is abundant and vulnerable to predation [24]. Therefore, predation patterns may be influenced by prey abundance and the habitat preferences of carnivorans and prey [59].

Since prey abundance was affected by habitat structure, distance from the burnt area perimeter and time-since-fire, the occurrence of carnivorans was also influenced indirectly by these factors. Subsequently, as we hypothesized, frequency of carnivoran was highest in the surroundings of the burnt area perimeter immediately after fire. Specifically, carnivoran occurrence maintained indirect relationships with distance from the burnt area perimeter via plant cover and small mammal abundance, while time-since-fire influenced the carnivoran community through its effect on plant cover. Fleshy fruit availability was also related to habitat structure and, consequently, to the distance from the burnt area perimeter. As an increasing distance from the burnt area perimeter corresponded to increasing fire severity, it can be inferred that fleshy fruit availability was also affected by fire severity. However, fruit resources do not seem to limit the presence of carnivorans in burnt areas. Instead, changes in the carnivoran occurrence could reflect the pace of vegetation recovery depending on fire severity.

Both, the carnivoran community and the red fox increased their frequency of occurrence inside the burnt area over time. Both may have benefited from the expansion of small mammals into the burnt area [19,21], since small mammal abundance was indirectly affected by both time and distance from the fire perimeter, mediated through its positive effect on plant cover. Our results are similar to those found in a study in Australia, where the red fox is an invasive species, and uses recently burnt forests and heathlands intensely [60,61]. Stone marten occurrence decreased away from the burnt area perimeter and refuges; however, it was affected neither by time nor time–distance interaction. A scarce plant cover may slow down the colonization of the stone marten. Although it avoids frequenting the same area as the red fox [62], and its diet overlaps with that of foxes [63], we did not find negative effects of the red fox on stone marten occurrence. Therefore, our results suggest that immediately after fire, the stone marten stayed around the burnt area perimeter, where refuges and food resources were more available [21]. Similarly, under predator control in Catalonia, fox abundance has increased due to its high reproductive rate and the decline of competing carnivorans, while stone marten abundance has strongly decreased and its subsequent recovery has started in areas of greater habitat complexity [64]. Although we found no clear relationship between fleshy fruit availability and carnivoran occurrence, fleshy fruit was, however, present in the GLMM models. It is likely that fruits are important for carnivorans in periods of low small mammal abundance [32,34]. During these periods, carnivorans can act as seed dispersers within burnt areas [16], and many feces containing seeds were observed in our study.

The rapid recolonization of rodents and the subsequent presence of carnivorous mammals after fire show the capacity of populations to recover from disturbances. However, the widespread management of recently burnt areas by post-fire salvage logging can influence plant regeneration, prey abundance [31], and, consequently, carnivoran presence. In this scenario, effort should be made to provide recommendations for more biodiversity-friendly harvesting practices [65] that allow the ecosystem services provided by mammals to recover in burnt and logged landscapes [66–68].

## 5. Conclusions

- Immediately after fire, carnivorans were more frequent close to the burnt area perimeter, where fire severity was low, and in places with greater small mammal abundance.
- Red fox occurrence in the burnt area surroundings increased with increasing small mammal abundance and plant cover, and with time-since-fire in the burnt area.
- Stone martens were found around the burnt area perimeter, probably because of their preference for high plant cover, and they were not significantly affected by small mammal abundance.

- The red fox apparently did not affect stone marten occurrence.
- Red fox and stone marten occurrence were not significantly related to fleshy fruit availability.
- Carnivoran responses to fire may be influenced, directly and indirectly, by habitat structure and resource availability.

**Supplementary Materials:** The following are available online at http://www.mdpi.com/1999-4907/11/8/855/s1, Figure S1: Principal component analysis of habitat structure, Table S1: Selected GLMM models and criteria, Text S1: Structural Equation Modelling (SEM).

**Author Contributions:** R.P.-G. and P.P. conceived and designed the research; R.P.-G. collected the data; R.P.-G. analyzed the data; R.P.-G. and P.P. wrote and reviewed the manuscript. All authors have read and agreed to the published version of the manuscript.

**Funding:** This study was funded by project CGL2014-54094-R from the Spanish Ministry of Economy and Competitiveness.

**Acknowledgments:** We thank C. Sanchez Cascante for her illustrations of Figure 1; G. Vila, P. Eijo, and the Galanthus Association for their help during this study. We also acknowledge the constructive comments of three anonymous reviewers. Part of this work is derived from the dataset generated during the doctoral thesis of Roger Puig-Gironès. We acknowledge the Spanish Ministry of Economy and Competitiveness for providing funds for this research.

**Conflicts of Interest:** The authors declare no conflicts of interest.

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
