# Peer review of "Mice and Habitat Complexity Attract Carnivorans to Recently Burnt Forests"

_forests, doi:10.3390/f11080855_

Round 1

Reviewer 1 Report

I have appended an annotated version of the MS with comments. In general the paper is interesting but there is a need to clarify the relationship between vegetation recovery and distance from the fire boundary. This should also take into account the stated relationship between fire intensity and distance from the fire boundary. 

I would like to recommend two papers:

Ecological separation between four species of carnivore in the western Pyrenees, Spain [microform] / June 1985 Hubertus Jacobus Van Hensbergen 

The second is a paper based on a three year study of small mammals before and after a controlled fire in a mediterranean area of South Africa. It is chapter 10. It details both the short and long term response of mammal communities to fire and relates this also to vegetation structure.  

Do Small Mammals Govern Vegetation Recovery After Fire in Fynbos? January 1992 DOI: 10.1007/978-3-642-76174-4_10 In book: Fire in South African Mountain Fynbos Publisher: Springer-Verlag

Hubertus Jacobus Van Hensbergen, S.A. Botha, G. G. Forsyth, David C Le Maitre

Author Response

English language and style 

Before sending the manuscript, a native professional English adviser reviewed the entire document. In addition, the reviewer does not indicate where we should apply extensive editing of English language and style required, but only indicates some individual words. Moreover reviewers 2 and 3 selected “English language and style are fine/minor spell check required”. Therefore, we believe that the paper should not go back to English correction. 

Comments and Suggestions for Authors 

I have appended an annotated version of the MS with comments. In general, the paper is interesting but there is a need to clarify the relationship between vegetation recovery and distance from the fire boundary. This should also take into account the stated relationship between fire intensity and distance from the fire boundary.  

We hope that the modifications following the reviewers' suggestions have clarified these relationships. 

Specific comments: 

Title: the correct English term is 'carnivores' or ' carnivore in the singular this needs to be corrected throughout the manuscript. 

The term CARNIVORAN is used in English when referring to the ORDER CARNIVORA (https://www.gbif.org/species/732), and may be synonymous of carnivore. The term is used in many specialized papers (for example: Plio-Pleistocene Carnivora of eastern Africa: species richness and turnover patterns (https://doi.org/10.1111/j.1096-3642.2005.00165.x); Morphological integration in the carnivoran skull (https://doi.org/10.1111/j.0014-3820.2006.tb01091.x); Déjà vu: the evolution of feeding morphologies in the Carnivora (https://doi.org/10.1093/icb/icm016); or A comparison of primate, carnivoran and rodent limb bone cross-sectional properties: are primates really unique? (https://doi.org/10.1006/jhev.2000.0420).  

On the other hand, in linguistic platforms such as 'Merriam-Webster', we find the following definitions: a. Definition of carnivoran (https://www.merriam-webster.com/dictionary/carnivoran): Any of an order (Carnivora) of placental mammals that possess bladelike carnassial teeth and feed primarily or exclusively on animal matter. b. Definition of carnivore (https://www.merriam-webster.com/dictionary/carnivore): (a) an animal (such as a dog, fox, crocodile, or shark) that feeds primarily or exclusively on animal matter: a carnivorous animal. (b) Any of an order (Carnivora) of animals that feed primarily or exclusively on animal matter: carnivoran

As suggested by our English adviser, we have incorporated "carnivorans (members of the mammalian order Carnivora)" to clarify when the term carnivoran first appears in the Abstract and Introduction. 

Line 102: lineal to line 

We have made the suggested change 

Line 104, 112 and 120: signals to sign 

As suggested by our English adviser, we now use the term ‘field signs’. See for example the paper: Methods of monitoring red foxes Vulpes vulpes and badgers Meles meles: Are field signs the answer?  In Mammal Review (2003). 

Line 135: 3 this is outside the area so should read -3 

Yes, that's right. 

Line 249: Figure 3 does not indicate any change for stone marten with time since fire since there is only one curve for stone marten and the timing of this curve is not indicated on the figure. 

The reviewer is right. However, the sentence refers to the carnivorous frequency of occurrence, not to the 'red fox' or the stone marten. The results for both species are presented in the following paragraphs. To improve this fact, we have incorporated letters to the graphs so that we use the following: Figure 3a; Figure 3b and Figure 3c. 

Line 273: Supplementary table S2 was not available with the manuscript. I believe that it should be included in the main paper since it is important evidence. 

We totally agree with the reviewer. Therefore, we have incorporated table S2, as table 3, in the text. At the same time, we have simplified this table to make it more readable for readers. 

Line 305: I am not sure if we can talk about the perception of predator risk affecting prey behaviour. I am not sure if prey are able to estimate predator risk. I believe it is more likely to be an innate avoidance of open areas. 

Perhaps, as the reviewer indicates, predator-prey evolution has led to innate avoidance of open areas so as not to be preyed upon. Individuals with more careful behaviour have probably been favoured by natural selection, transmitting this character between generations. However, we cannot know whether the prey (rodents in our case) perceive the predator risk or not. Other aspects such as nocturnal activity and avoidance of full moon nights (to evade being seen by predators) suggest that they may perceive the danger. 

 We have modified the sentence in accordance with these thoughts to  

Accordingly, post-fire habitat structure and, likely, the perception of predator risk affects prey behaviour” 

Line 306: The statement about drawing predators into the area does not relate to perceptions of predator risk. This sentence should be split into two. 

Following reviewer’s suggestion, we have split the sentence so that: 

“[…] affect prey behaviour [29,30]. Moreover, habitat structure draw predators […]” 

Line 317: It is worth more than mentioning, it seems likely that the gradient in fruit availability with distance from the edge is caused directly by the similar gradient in fire intensity and that the changes in carnivore distribution post fire reflect the pace of vegetation recovery following different fire intensity. 

We believe, as the reviewer points out, that it is sensible to incorporate this type of message here, especially so that readers remember that distance is a surrogate for severity and what this implies in our results. That is why we have incorporated the following sentence: 

"As an increasing distance from the burnt area perimeter corresponded to increasing fire severity, it can be inferred that fleshy fruit availability was also affected by fire severity. However, fruit resources do not seem to limit the presence of carnivorans in burnt areas. Instead, changes in the carnivoran occurrence could reflect the pace of vegetation recovery depending on fire severity

Figure 2: This should read 'plant cover' in both parts of the figure 

We have made the suggested change 

I would expect there to be a biologically meaningful relationship between small mammal abundance and fleshy fruit availability. Fleshy fruit might attract small mammals and small mammals may reduce fleshy fruit availability. 

In our previous work (Puig-Gironès et al. 2018 and Puig-Gironès et al. 2020) we saw that the availability of seeds and trophic resources (seeds and fleshy fruit) did not significantly affect rodents' recolonization in the first months after the fire. Therefore, we decided not to incorporate this relationship in the Structural Equation Models (SEM), in order to simplify this model, which is already very complex. 

Figure 3: This graphic needs to have an explanation of the distance variable involved. 

We agree with the reviewer, so we have incorporated the following sentence in the figure legend:  

For the distance variable, negative values were assigned to stations in the unburnt area (-1= 1 to 50 m; -2= 51 to 100 m; and -3= more than -101 m from the burnt area perimeter), whereas positive values indicate distances within the burnt area (0= 0 m; 1= 1 to 50 m; 2= 51 to 100 m; 3= 101 to 200 m; 4= 201 to 400 m; 5= 401 to 800 m; and 6= more than 801 m from the burnt area perimeter)” 

 Table 2: Explicative to explanatory 

We have made the suggested change 

I would like to recommend two papers:  

Ecological separation between four species of carnivore in the western Pyrenees, Spain [microform] / June 1985 Hubertus Jacobus Van Hensbergen   

The second is a paper based on a three year study of small mammals before and after a controlled fire in a mediterranean area of South Africa. It is chapter 10. It details both the short and long term response of mammal communities to fire and relates this also to vegetation structure.   

Do Small Mammals Govern Vegetation Recovery After Fire in Fynbos? January 1992 DOI: 10.1007/978-3-642-76174-4_10 In book: Fire in South African Mountain Fynbos Publisher: Springer-Verlag 

Hubertus Jacobus Van Hensbergen, S.A. Botha, G. G. Forsyth, David C Le Maitre 

Thanks for the bibliographic recommendations, we have read them carefully. Since other reviewers have suggested increasing the literature on carnivore diets, we have ruled out increasing the literature on topics that are already sufficiently supported by the literature.  

On the other hand, one of the suggested reference is a manuscript that has not been published and is not citable. 

Reviewer 2 Report

Brief summary:

The aim of the paper was to evaluate the influence of fire on carnivoran community in Mediterranean forests of Spain. In particular, the Authors were interested, how different environmental variables, including e.g. habitat structure, fire severity or food resource availability, influenced the carnivoran frequency of occurrence. Addressing that question is important since it contributes to expanding our knowledge on the complex effects of fire disturbance on ecosystems, including animal communities. Furthermore, the results of the paper highlight the potential of animal populations to recover after wildfires and suggest that post-fire management in form of salvage logging may eventually negatively affect those processes.

Broad comments:

The paper presents sound analysis of empirical data from Mediterranean Europe in a well-structured way. The results should be interesting for a broad audience of both fire and animal ecologists. The article is well-written and easy to understand. Besides one or two typos I was not able to find any section of the article which would require English editing. After throughout Introduction, the Authors have formulated three clear hypotheses, addressed in the statistical analyses. However, the paper could profit from mentioning the hypotheses again in the Results and/or in the Discussion. Since post-fire management is mentioned in the Discussion, it would be valuable to expand that issue. In the form it is presented now it seems a bit ‘unfinished’, especially the last sentence. Maybe some broader context would be beneficial in the final section of the article before Conclusions.

Specific comments:

Line 61: I would add fire severity (at least) to the list of factors influencing post-fire plant recovery.

Line 76: What does “severely burnt areas” mean precisely? It is a general term, not specific. Maybe this is included in the site description offered by the cited references (Line 86), however, it would be valuable to specify fire severity in each of the study sites.

Lines 81, 83, 85: Readers may wonder why one of the sites was sampled 18 months after fire. Any explanation?

Line 133: A reference for the given statement on carnivoran ecology?

Lines 182-183: In my opinion, Readers would benefit from at least short description of the five fire severity classes.

Lines 216-218: I agree that this may be fully justified in the analyzed case. However, are there any data supporting that this is a general rule? Maybe it is worth to address that issue in the Discussion?

Lines 315-317: See above (Lines 216-218). Again, it may be true for the analyzed case/analyzed burns/analyzed fires. But is it always like this? In my opinion, it would be worth adding here some wider context with data from other burns as well.

Lines 333-334: Any reference supporting the statement that fruits are important for carnivorans in periods of low small mammal abundance?

Lines 527-530: Did I miss those two references [66, 67] in the article, or are they actually missing?

Author Response

Comments and Suggestions for Authors 

Brief summary: 

The aim of the paper was to evaluate the influence of fire on carnivoran community in Mediterranean forests of Spain. In particular, the Authors were interested, how different environmental variables, including e.g. habitat structure, fire severity or food resource availability, influenced the carnivoran frequency of occurrence. Addressing that question is important since it contributes to expanding our knowledge on the complex effects of fire disturbance on ecosystems, including animal communities. Furthermore, the results of the paper highlight the potential of animal populations to recover after wildfires and suggest that post-fire management in form of salvage logging may eventually negatively affect those processes. 

Broad comments: 

The paper presents sound analysis of empirical data from Mediterranean Europe in a well-structured way. The results should be interesting for a broad audience of both fire and animal ecologists. The article is well-written and easy to understand. Besides one or two typos I was not able to find any section of the article which would require English editing. After throughout Introduction, the Authors have formulated three clear hypotheses, addressed in the statistical analyses.  

Thanks for the comments 

However, the paper could profit from mentioning the hypotheses again in the Results and/or in the Discussion. 

We have included the different phrases in the Discussion: 

“Then, the results partially support the hypothesis that small mammal abundance and fleshy fruit availability affect carnivoran abundance in recently burnt areas”. 

“Subsequently, as we hypothetized, frequency of carnivoran was highest in the surroundings of the burnt area perimeter immediately after fire”. 

On the other hand, we understand that the phrase:  

"Therefore, our results suggest that, immediately after fire, the stone marten stayed around the burnt area perimeter, where refuges and food resources were more available [21]" can already do this function requested by the reviewer. 

Since post-fire management is mentioned in the Discussion, it would be valuable to expand that issue. In the form it is presented now it seems a bit ‘unfinished’, especially the last sentence. Maybe some broader context would be beneficial in the final section of the article before Conclusions. 

We have tried to conclude the discussion by rewriting the paragraph as follows: 

 "The rapid recolonization of rodents and the subsequent presence of carnivorous mammals after fire show the capacity of populations to recover from disturbances. However, the widespread management of recently burnt areas by post-fire salvage logging, can influence plant regeneration, prey abundance [31], and, consequently, carnivoran presence. In this scenario, an effort should be made to provide recommendations for more biodiversity-friendly harvesting practices [65] that allow the ecosystem services provided by mammals to recover in burnt and logged landscapes [e.g. 66-68]" 

Specific comments: 

Line 61: I would add fire severity (at least) to the list of factors influencing post-fire plant recovery. 

As the reviewer suggests, we have added the fire severity to the list. 

 Line 76: What does “severely burnt areas” mean precisely? It is a general term, not specific. Maybe this is included in the site description offered by the cited references (Line 86), however, it would be valuable to specify fire severity in each of the study sites. 

To clarify this concept, we have incorporated the following sentence: 

“[...] severely burnt areas (i.e. >80% of immediate tree mortality within the fire perimeter) [...]”  

Lines 81, 83, 85: Readers may wonder why one of the sites was sampled 18 months after fire. Any explanation? 

Sorry, this was not 18 months but 18 weeks after fire. There were temporal differences in sampling between sites because we could not decide the fire dates and tried to have a suitable site replication (recently burnt area that had been salvage logged). Rasquera was sampled from 18 weeks after fire because the fire was occurred 12 and 16 weeks before La Jonquera and Ger fires, respectively.  

While we believe readers will understand the difficulty of sampling for "naturally occurring" fires at the same time, we have included the following phrase:  

"because it occurred 12 and 16 weeks before La Jonquera and Ger fires, respectively” 

Line 133: A reference for the given statement on carnivoran ecology? 

As requested by the reviewer, we have incorporated the reference [9] in this sentence. 

Lines 182-183: In my opinion, Readers would benefit from at least short description of the five fire severity classes. 

To give the reader more information we have incorporated the following sentence: 

“[…] using a categorical scale following the methodology described by Keeley [45], were: (1) unburned; (2) unburned but plants exhibit leaf loss from radiated heat; (3) canopy trees with green needles and herbs charred; (4) moderate or severe surface burn; and (5) deep burning or crown fire”.   

Lines 216-218: I agree that this may be fully justified in the analyzed case. However, are there any data supporting that this is a general rule? Maybe it is worth to address that issue in the Discussion? 

In the previous exploration of our data (section 1 of the hierarchical process used) we saw that both were correlated. This happened with our data, we cannot guarantee that it is a standard of all fires, although we can imagine that it is.  

To reduce the perception of a norm, we have included the following sentence:  

"After performing multicollinearity diagnostics and since distance from the burnt area perimeter and fire severity were correlated, we eliminated [....]". 

Lines 315-317: See above (Lines 216-218). Again, it may be true for the analyzed case/analyzed burns/analyzed fires. But is it always like this? In my opinion, it would be worth adding here some wider context with data from other burns as well. 

Although we understand the reviewer's scientific interest in knowing the generalization of this fact in other disturbed areas, we do not consider that we have to do this research exercise or divert the paper's interest towards this fact. Since we are only talking about our specific case, for which the severity of the fire was correlated with the distance to the perimeter, we believe that it is understood when we say:  

"[...] in our analyses the distance from the burnt area perimeter also correspond to a fire severity gradient, [...]": 

However, at the suggestion of reviewer 1, we have incorporated the following phrase: 

"As an increasing distance from the burnt area perimeter corresponded to increasing fire severity, it can be inferred that fleshy fruit availability was also affected by fire severity. However, fruit resources do not seem to limit the presence of carnivorans in burnt areas. Instead, changes in the carnivoran occurrence could reflect the pace of vegetation recovery depending on fire severity

Lines 333-334: Any reference supporting the statement that fruits are important for carnivorans in periods of low small mammal abundance? 

Bibliographic references [32, 34] have been incorporated in the text. 

Lines 527-530: Did I miss those two references [66, 67] in the article, or are they actually missing? 

Both references are found in the supplementary material. To avoid this problem, both references have been removed from the references section of the main text, and we have left it only in the supplementary material. 

Reviewer 3 Report

The interactions among predators, prey, habitat and disturbances such as wildfire are the subject of much investigation around the globe. The current manuscript extends this theme to the forested landscapes of Spain. While not unique in scope or findings, the study reaffirms the importance of prey abundance and habitat complexity in influencing patterns in the occurrence of predators such as the red fox immediately after as well as following fire.

For the most part the manuscript is well written, and the science upon which it is based, sound. That said, it would benefit from some additional explanatory detail, particularly in the methods section of the text. To that end I offer the following suggested editorial changes:

Throughout the text the terms "predator(s)", "carnivore(s)" and "carnivoran" (the latter mostly used) are interchanged. To the extent possible standardise and use one term only - carnivore(s). This would make it consistent with the broader global literature.

ABSTRACT

Lines 7-8. Sentence "Despite the importance of large predators in the reconstitution of terrestrial trophic webs, still little is known about how fire affects carnivores". This sentence seems a little incongruous given that your findings indicate just how strongly the occurrence of carnivores is influenced by the abundance of key prey and associated with that, habitat complexity. Thus, carnivores are as much a part of the bigger whole as they are a driver. That is the main message so reword the sentence accordingly.

Lines 23-25. The sentence about harvesting impacts seems a little out of place given that you didn't look at that directly - consider deleting.

INTRODUCTION

Line 69. Delete "Accordingly" and just simply state "We predicted that ....".

METHODS

Lines 101-110. Here you specify that survey transects for carnivore sign were 900 m x 10 m, yet you indicate a search area 3 m either side of the mid-line of the transect. Why wasn't the outer 2 m either side of the 3 m mark also sampled? In which case the transect is actually 900 m x 6 m. Similarly, no information is provided about how much of the 900 m transect was situated outside of the burned area, and how much within it, for each transect across each of the three study sites. Can you please specify?

More detail needs to be provided here about how independent, or otherwise, transects were from one another. Transects were a median distance of 250 m apart - this seems to be very close given the movement capabilities of foxes and martens. Can you add some further sentences about this and the consequences for later statistical analysis, if any?

Line 108. Stated that "an observer" recorded carnivore sign along each transect. Was the observer the same for all transects or were multiple observers used? If the latter, how was consistency in search effort maintained?

Lines 110-111. Here it is stated that faeces of carnivores were removed from a transect when they were encountered, to prevent double counting. Given that carnivores can be territorial, what was the potential impact of removing scats, instead of leaving them and marking them in a way that retained their function on-site? Put another way, was their any evidence in the data that showed a negative effect of your actions?

RESULTS

Table 1. Stats need to be added on how many transects across each study area were found to have evidence of carnivores and, furthermore, how even or skewed the evidence was (i.e. were the scats evenly distributed or mostly occurred at some sites?).

Lines 135-139. Why were the segment lengths along transects non-standard (i.e. 50 m, 100 m up to 400 m)? Was there a pattern in the evidence that indicated that was an appropriate thing to do, or otherwise justifiable?

Line 212. Replace "weight" with "magnitude".

Figure 2. "Plan Cover" should read "Plant Cover".

Line 250. Here you state that small mammal abundance had recovered at the fire edge. Did you have any evidence that it had in fact declined in the first place?

Figure 3. Why do the y-axes extend below zero into negative values, when probability of occurrence can only be a value between 0 and 1.

Table 2. "Explicative variables" should be "Explanatory variables".

Lines 289-292. Both height of vegetation and red fox occurrence were non-significant in explaining stone marten occurrence. Change the text accordingly.

DISCUSSION

Lines 294-298. Isn't this just a reflection of food preferences (i.e. small mammals over fruit)? This section of text would benefit by broader discussion about the diets of foxes and martens, using published literature.

Lines 297-300. Make it clear here that you are referring to a related study, the findings from which may help explain the trends observed in your present study.

Lines 300-301. Can you further explain why small mammal abundance didn't come out as an important explanatory variable for the stone marten? Further discussion is warranted here of reasons why that was the case.

Lines 307-308. Delete this sentence as it just repeats what you have said earlier.

Lines 313-315. How much do fleshy fruits contribute to the diet of foxes and martens? Expand the text here using existing literature.

Lines 337-343. This section on impacts of harvesting and the importance of CWD seems a little out of place given the focus on prey availability and habitat complexity. Perhaps better relate the CWD part of the story by nominating that it forms part of habitat complexity. Or otherwise delete.

Author Response

Comments and Suggestions for Authors 

The interactions among predators, prey, habitat and disturbances such as wildfire are the subject of much investigation around the globe. The current manuscript extends this theme to the forested landscapes of Spain. While not unique in scope or findings, the study reaffirms the importance of prey abundance and habitat complexity in influencing patterns in the occurrence of predators such as the red fox immediately after as well as following fire. 

For the most part the manuscript is well written, and the science upon which it is based, sound. That said, it would benefit from some additional explanatory detail, particularly in the methods section of the text.  

We hope that the modifications following the reviewers' suggestions have clarified these relationships. 

To that end I offer the following suggested editorial changes: Throughout the text the terms "predator(s)", "carnivore(s)" and "carnivoran" (the latter mostly used) are interchanged. To the extent possible standardise and use one term only - carnivore(s). This would make it consistent with the broader global literature. 

Thanks for the suggestion, however we only use the term carnivore once and as a descriptive term.  

We make little use of the concept predator in those contexts where reference is made to the predator-prey relationship. 

On the other hand, the term CARNIVORAN is used in English when referring to the ORDER CARNIVORA (https://www.gbif.org/species/732), and may be synonymous of carnivore. The term is used in many specialized papers (for example: Plio-Pleistocene Carnivora of eastern Africa: species richness and turnover patterns (https://doi.org/10.1111/j.1096-3642.2005.00165.x); Morphological integration in the carnivoran skull (https://doi.org/10.1111/j.0014-3820.2006.tb01091.x); Déjà vu: the evolution of feeding morphologies in the Carnivora (https://doi.org/10.1093/icb/icm016); or A comparison of primate, carnivoran and rodent limb bone cross-sectional properties: are primates really unique? (https://doi.org/10.1006/jhev.2000.0420).  

On the other hand, in linguistic platforms such as 'Merriam-Webster', we find the following definitions: a. Definition of carnivoran (https://www.merriam-webster.com/dictionary/carnivoran): Any of an order (Carnivora) of placental mammals that possess bladelike carnassial teeth and feed primarily or exclusively on animal matter. b. Definition of carnivore (https://www.merriam-webster.com/dictionary/carnivore): (a) an animal (such as a dog, fox, crocodile, or shark) that feeds primarily or exclusively on animal matter: a carnivorous animal. (b) Any of an order (Carnivora) of animals that feed primarily or exclusively on animal matter: carnivoran

As suggested by our English adviser, we have incorporated "carnivorans (members of the mammalian order Carnivora)" to clarify when the term carnivoran first appears in the Abstract and Introduction. 

Specific comments:

ABSTRACT

Lines 7-8. Sentence "Despite the importance of large predators in the reconstitution of terrestrial trophic webs, still little is known about how fire affects carnivores". This sentence seems a little incongruous given that your findings indicate just how strongly the occurrence of carnivores is influenced by the abundance of key prey and associated with that, habitat complexity. Thus, carnivores are as much a part of the bigger whole as they are a driver. That is the main message so reword the sentence accordingly.

We have eliminated the text that generates inconsistency, rewriting the phrase as follows:

"Despite the importance of large predators in terrestrial trophic webs, still little is known about how fire affects carnivorans (the mammalian order Carnivora)”

Lines 23-25. The sentence about harvesting impacts seems a little out of place given that you didn't look at that directly - consider deleting.

This part of the summary is certainly not in line with the previous statement, and seems to be detached from the rest of the summary. That is why we have eliminated this phrase

INTRODUCTION

Line 69. Delete "Accordingly" and just simply state "We predicted that ....".

We have made the suggested change

METHODS

Lines 101-110. Here you specify that survey transects for carnivore sign were 900 m x 10 m, yet you indicate a search area 3 m either side of the mid-line of the transect. Why wasn't the outer 2 m either side of the 3 m mark also sampled? In which case the transect is actually 900 m x 6 m. Similarly, no information is provided about how much of the 900 m transect was situated outside of the burned area, and how much within it, for each transect across each of the three study sites. Can you please specify?

Thanks for this comment, since the 10-metre width was a writing error.

The specific information of the transect (how much surface is inside and outside the burnt area), is explained in the next paragraph about the treatment of variables. In this first paragraph, we explain the method used, and it is convenient to include a short explanation:

"Transects had approximately 100 m in the unburnt area and were separated by at least 50 m [...]".

More detail needs to be provided here about how independent, or otherwise, transects were from one another. Transects were a median distance of 250 m apart - this seems to be very close given the movement capabilities of foxes and martens. Can you add some further sentences about this and the consequences for later statistical analysis, if any?

The reviewer is right. The median distance of 250m corresponds to complete set of transect used in the global project. However, for this article, not all transects were sampled and the distance has to be recalculated (we forgot to do it for this manuscript). Once recalculated, the minimum, distances between adjacent transects are longer than 500 m in 20 cases. For the 13 contiguous transects that are closer than 500 meters at any point of the transects, the table and plot (adjunt document) below show minimum, maximum and average distances. 

Transect 

Distance (m) max 

Distance (m) min 

Average 

Ger 2 to 3 

459 

141 

300 

Rasquera 2 to 3 

517 

209 

363 

Ger 0 to 1 

627 

154 

390.5 

APV 1 to 2 

650 

157 

403.5 

Ger 4 to 5 

650 

191 

420.5 

Rasquera 0 to 1 

769 

142 

455.5 

Rasquera 4 to 5 

1047 

162 

604.5 

Massarac 1 to 2 

1060 

197 

628.5 

MasBrugat 1 to 2 

1110 

205 

657.5 

Cantallops 1 to 2 

1021 

381 

701 

SantaLlucia 1 to 2 

1236 

225 

730.5 

SantJulia 1 to 2 

1305 

489 

897 

FontBonaigua 1 to 2 

2700 

384 

1542 

Average 

1011.6 

233.6 

622.6 

The density and abundance of foxes are highly variable depending on the quality of the habitat, the quantity and type of food, the time of year, the presence of other competing species, the presence of diseases, population management measures (hunting and control) and the interaction of all these factors. In the United Kingdom, densities have been estimated at between 0.49 and 2.95 foxes/km2. In France between 0.23 and 3.81 foxes/km2. In Spain densities and abundances have been estimated between 0.8 foxes/km2 in dryland areas and 2.5 foxes/km2 in irrigated areas (see López-Martín & Salvador 2017. Zorro – Vulpes vulpes Linnaeus, 1758 in Enciclopedia Virtual de los Vertebrados Españoles. http://dx.doi.org/10.20350/digitalCSIC/8867; Díaz-Ruiz et al. 2016. Drivers of red fox (Vulpes vulpes) daily activity: prey availability, human disturbance or habitat structure? Journal of Zoology 298(2): 128-138; Blanco 1986. On the diet, size and use of home range and activity patterns of a red fox In Central Spain. Acta theriologica 31:547-552. doi: 10.4098/AT.arch.86-48). 

With these data in mind, we think that the distance between transects is sufficient to discriminate between territories (knowing the existence of overlaps and invasions to neighbouring territories). As the probability of finding field signs from the same individual in two contiguous transects of 6 meters wide is very low, we think that our transects can be considered as spatially independent and that no changes should be made to the data analyses. 

 We have therefore changed the text as follows: 

“Transects had approximately 100 m in the unburnt area. Adjacent transects were separated by at least 140 m in its closest point, although 61% of adjacent transects were separated by at least 500 m. The average distance between adjacent transects of 2048 m, ensured a reasonable spatial independence of transects for carnivorans.” 

Line 108. Stated that "an observer" recorded carnivore sign along each transect. Was the observer the same for all transects or were multiple observers used? If the latter, how was consistency in search effort maintained?

The observer was the same for all transects and study areas.

Lines 110-111. Here it is stated that faeces of carnivores were removed from a transect when they were encountered, to prevent double counting. Given that carnivores can be territorial, what was the potential impact of removing scats, instead of leaving them and marking them in a way that retained their function on-site? Put another way, was their any evidence in the data that showed a negative effect of your actions?

The faeces were removed from the transects (6 meters wide), not from the study area. The transect area (around 5.4 ha in total) is small when compared to the home range or territory of a carnivoran, and the number of faeces removed from a transect ranged from 2 to 21. Taking this into account, we do not believe that our sampling had any negative effect on carnivoran behaviour.

RESULTS

Table 1. Stats need to be added on how many transects across each study area were found to have evidence of carnivores and, furthermore, how even or skewed the evidence was (i.e. were the scats evenly distributed or mostly occurred at some sites?).

To address this comment, we have introduced two sentences, one in the M&Ms and one in the results:

“La Jonquera concentrated 22 of the 33 transects analysed, while Ger had 6 and Rasquera 5”

“All transects had field signs (range 2-21 for the duration of the study), although four transects concentrated 41% of the field signs found"

Lines 135-139. Why were the segment lengths along transects non-standard (i.e. 50 m, 100 m up to 400 m)? Was there a pattern in the evidence that indicated that was an appropriate thing to do, or otherwise justifiable?

Initially, we wanted to locate stations at standard distances from the fire perimeter. However, we set up our transects before a map of the burned area was available. Although we tried to locate stations along transects at a geometric progression of distances (0-50-100-200-400-800-1600) this was not totally possible because fire perimeters were irregular. Once we had maps, we calculated the exact position of stations used in analyses and for graphical representations.

Line 212. Replace "weight" with "magnitude".

We have made the suggested change

Figure 2. "Plan Cover" should read "Plant Cover".

We have made the suggested change

Line 250. Here you state that small mammal abundance had recovered at the fire edge. Did you have any evidence that it had in fact declined in the first place?

The evidence is the comparison in the number of small mammals captures between the burnt and the unburnt stations during the initial and final samplings, for example:

  • 0 – 10 weeks after fire: we found 0.3/1 wood mouse proportion of captures in unburnt areas and 0.15/1 in 25 m from burn area perimeter
  • 90 to 120 weeks after fire: we found 0.4/1 wood mouse proportion of captures in unburnt areas and 0.5/1 in 25 m from burn area perimeter

(see Puig-Gironès et al 2018 IJWF for more information)

Figure 3. Why do the y-axes extend below zero into negative values, when probability of occurrence can only be a value between 0 and 1.

Although the modelled frequency of occurrence goes, roughly, from 0 to 1, the error can go further, and this is why the y-axes has negative labels. Thanks to this comment, we saw that the last two graphics of the red fox were badly placed.

Table 2. "Explicative variables" should be "Explanatory variables".

We have made the suggested change

Lines 289-292. Both height of vegetation and red fox occurrence were non-significant in explaining stone marten occurrence. Change the text accordingly.

Although in the GLMM models variable significance may not be the main element to evaluate the different variables of the model, but the weight of the variable in the Akaike Information Criteria weight (AICω+) is more important, we have modified the sentence to make it clearer:

“Abundance of small mammals, height of vegetation, and red fox occurrence also had a positive influence on stone marten occurrence (although not significant, they were present in the selected model)”

DISCUSSION

Lines 294-298. Isn't this just a reflection of food preferences (i.e. small mammals over fruit)? This section of text would benefit by broader discussion about the diets of foxes and martens, using published literature.

In our previous work (Puig-Gironès et al. 2018 and Puig-Gironès et al. 2020) we saw that the availability of seeds and trophic resources (seeds and fleshy fruit) did not significantly affect rodents' recolonization in the first months after the fire. We have now added the following sentence:

"In burnt areas distributed throughout Catalonia, we reported that rodent foraging activity increased considerably within the burnt area 15 months after fire, regardless on the fleshy fruit and seeds availability [21; 31]."

Regarding the topic of carnivore diet we have added the following sentence and references:

“Similarly, red fox increased its presence in sites with greater small mammal abundance and was not significantly affected by fleshy fruit availability, in accordance with its usual relevance (approximately 15% of frequency of occurrence on average in Spain) in the diet of the species [32, 55]”.

[55] Delibes-Mateos, M.; De Simon, J.F.; Villafuerte, R.; Ferreras, P. Feeding responses of the red fox (Vulpes vulpes) to different wild rabbit (Oryctolagus cuniculus) densities: a regional approach. Eur J Wildl Res 2008, 54, 71-78, doi:10.1007/s10344-007-0111-5.

Lines 297-300. Make it clear here that you are referring to a related study, the findings from which may help explain the trends observed in your present study.

To clarify this, we modify the sentence as follows:

“In burnt areas distributed throughout Catalonia, we previously reported that rodent foraging activity increased considerably within the burnt area [...]”.

Lines 300-301. Can you further explain why small mammal abundance didn't come out as an important explanatory variable for the stone marten? Further discussion is warranted here of reasons why that was the case.

We've tried to explain why small mammal abundance didn't come out as an important explanatory variable for the stone marten with the following sentence and bibliography:

“[…] stone marten occurrence was not significantly affected by small mammals, probably due to its rich diet in fruits and insects [34, 56, 57]

[56] References: Ruiz-Olmo, J.; Palazon, S. Diet of the stone marten (Martes foina Erxleben, 1777) in northeastern Spain. Acta Vertebr 1993, 20, 59-67

[57] Posłuszny, M.; Pilot, M.; Goszczyński, J.; Gralak, B. Diet of sympatric pine marten (Martes martes) and stone marten (Martes foina) identified by genotyping of DNA from faeces. Ann Zool Fennici 2007, 44, 269-284

Lines 307-308. Delete this sentence as it just repeats what you have said earlier.

We believe that with the recent changes made, this phrase retains its interest as a summary of possible predation patterns.

Lines 313-315. How much do fleshy fruits contribute to the diet of foxes and martens? Expand the text here using existing literature.

This has been answered in the previous questions (above).

Lines 337-343. This section on impacts of harvesting and the importance of CWD seems a little out of place given the focus on prey availability and habitat complexity. Perhaps better relate the CWD part of the story by nominating that it forms part of habitat complexity. Or otherwise delete.

We have tried to conclude the discussion by rewriting the paragraph as follows:

"The rapid recolonization of rodents and the subsequent presence of carnivorous mammals after fire show the capacity of populations to recover from disturbances. However, the widespread management of recently burnt areas by post-fire salvage logging, can influence plant regeneration, prey abundance [31], and, consequently, carnivoran presence. In this scenario, an effort should be made to provide recommendations for more biodiversity-friendly harvesting practices [65] that allow the ecosystem services provided by mammals to recover in burnt and logged landscapes [e.g. 66-68]"

Round 2

Reviewer 3 Report

Thank you for addressing each of my queries, in full. The resultant revised manuscript reads well.